# Construction of ssDNA-Attached LR-Chimera Involving Z-DNA for ZBP1 Binding Analysis

**DOI:** 10.3390/molecules27123706

**Published:** 2022-06-09

**Authors:** Lin Li, Ran An, Xingguo Liang

**Affiliations:** 1College of Food Science and Engineering, Ocean University of China, Qingdao 266003, China; xbnllilin@163.com; 2Laboratory for Marine Drugs and Bioproducts, Qingdao National Laboratory for Marine Science and Technology, Qingdao 266235, China

**Keywords:** Z-DNA, binding constant, ZBP1, circular DNA, bio-layer interferometry

## Abstract

The binding of proteins to Z-DNA is hard to analyze, especially for short non-modified DNA, because it is easily transferred to B-DNA. Here, by the hybridization of a larger circular single-stranded DNA (ssDNA) with a smaller one, an LR-chimera (involving a left-handed part and a right-handed one) with an ssDNA loop is produced. The circular ssDNAs are prepared by the hybridization of two ssDNA fragments to form two nicks, followed by nick sealing with T4 DNA ligase. No splint (a scaffold DNA for circularizing ssDNA) is required, and no polymeric byproducts are produced. The ssDNA loop on the LR-chimera can be used to attach it with other molecules by hybridization with another ssDNA. The gel shift binding assay with Z-DNA specific binding antibody (Z22) or Z-DNA binding protein 1 (ZBP1) shows that stable Z-DNA can form under physiological ionic conditions even when the extra ssDNA part is present. Concretely, a 5′-terminal biotin-modified DNA oligonucleotide complementary to the ssDNA loop on the LR-chimera is used to attach it on the surface of a biosensor inlaid with streptavidin molecules, and the binding constant of ZBP1 with Z-DNA is analyzed by BLI (bio-layer interferometry). This approach is convenient for quantitatively analyzing the binding dynamics of Z-DNA with other molecules.

## 1. Introduction

Usually, research on Z-DNA, one of the basic and very interesting secondary DNA structures with left-handed helix and the zig-zag phosphate skeleton, is elusive and controversial [1,2]. Much evidence shows that the Z-DNA of alternative pyrimidine purine (APP) sequences really exists in vivo and is associated with diverse diseases [3,4,5,6,7]. It has been shown that Z-DNA is involved in regulating gene expression, inducing chromosome recombination, and so on [8,9,10,11,12,13,14,15,16]. However, Z-DNA is hard to form without topological constraint or special conditions such as chemical modification or high salt concentrations because it is easily transferred to B-DNA. As a result, Z-DNA research is much more difficult compared with B-DNA research. For example, the binding dynamics between Z-DNA and Z-DNA binding proteins, which is essential to understand the biological function of Z-DNA in vivo, have not been comprehensively investigated, because either long plasmid DNAs with negative supercoil or short modified DNAs are needed. Long DNA may bring some interfering factors, such as uncertainty of the position of Z-DNA, non-specific binding, and so on. Chemical modification may not reflect the real binding dynamics of unmodified DNA.

The mystery of the interaction of Z-DNA with other molecules is far from being revealed, although some characteristics and parameters for Z-DNA binding proteins have been tackled to some extent. Runkel and Nordheim used the d(CG)_11_- and d(CG)_16_-inserted plasmids to study the binding site and their affinity to the Z-DNA-specific binding antibody (Z22) by foot-printing using diethylpyrocarbonate [17]. Bromine-modified d(CG)_n_ is commonly used to measure the affinity constants of Z-DNA with adenosine deaminase acting on RNA (ADAR1), Z-DNA binding protein 1 (ZBP1), and poxvirus E3L by using surface plasma resonance (SPR) or bio-impedance analysis (BIA), because it takes Z-DNA conformation under physiological ionic conditions [18,19,20,21]. Utilization of non-modified Z-DNA for the quantitative analysis of Z-DNA binding affinity is difficult because the plasmid DNA is too big and hard to attach to the surface. For modified Z-DNA, on the other hand, it is different from Z-DNA in vivo with non-modified bases. Obviously, a facile approach using stable Z-DNA without chemical modification is required to obtain affinity parameters under various conditions. There is an urgent need to construct a system for evaluating the affinity of Z-DNA binding proteins especially under physiological ionic conditions.

If the bio-layer interferometry (BLI), a real-time dip-and-read assay system to study biomolecular interactions, can be used to analyze Z-DNA of relatively short non-modified DNAs, it is very helpful for Z-DNA affinity analysis. The binding information is obtained by monitoring the change of light interference with time (interference spectrum) to reflect the binding dynamics between biomolecules [22,23]. BLI has been used to study the protein–protein interaction [24,25,26], as well as protein–nucleic acid interaction [27,28,29]. Obviously, construction of stable Z-DNA, which can be attached on the BLI probe, is the key technique to reach this goal. Recently, we have successfully prepared stable non-modified Z-DNA (LR-chimera involving the left-handed part and the right-handed one) under physiological ionic conditions (e.g., 10 mM MgCl_2_) using two complementary 74~111 bp long circular single-stranded DNA (ssDNA) [30]. The circularization of ssDNA is a critical step, which needs to be improved [31,32,33,34]. In addition, the LR-chimera we prepared is difficult to attach onto the surface of the probe, because it has no ssDNA part for further hybridization. This inspired us to prepare LR-chimeras with an ssDNA loop for attaching it to the BLI probe (Figure 1).

In this study, we developed a new approach to efficiently prepare circular ssDNAs without using splints (an extra short ssDNA to draw close two ends of the precursor ssDNA to form a nick) [34]. After hybridizing two DNA fragments to form two nicks, circular ssDNA can be obtained by nick sealing with T4 DNA ligase (Figure 2A). The LR-chimera that attaches a single-stranded part for hybridization with other ssDNA was obtained by hybridization of a larger circular ssDNA and a smaller one (Figure 1). A 5′-terminal biotin-modified oligonucleotide could hybridize with the LR-chimera and attach it on the surface of a biosensor for the binding analysis of ZBP1 (its binding to Z-DNA/Z-RNA is closely related to immunity) [35,36,37,38]. This method is simple to operate and can realize the quantitative analysis of Z-DNA binding under a variety of ion conditions.

## 2. Results and Discussion

### 2.1. Preparation of Circular ssDNA by Two DNA Hairpins

The preparation of an LR-chimera for comprehensive analysis requires a relatively large amount of circular ssDNAs. For efficient preparation of an LR-chimera consisting of two circular ssDNAs, we first developed a new circularization approach to completely suppress the polymeric byproducts (Figure 2). At present, researchers have significantly improved the circularization efficiency by reducing MgCl_2_ concentration [31], using the terminal hairpin of the ssDNA substrate [32] or frozen/lyophilization/cyclization (FLC) [33]. Sui et al. reported a one-pot ligation method to efficiently prepare longer circular ssDNA (>90 nt) [34], but several splints were needed, and the yield was not high enough in some cases. In our new approach presented here, each ssDNA circle was synthesized by using two fragments, which could hybridize with each other to form two nicks (Figure 2A). After ligation by T4 DNA ligase, a circular ssDNA with two hairpin-like structures could be formed. As shown in Figure 2B, for efficient ligation, a 5~7 bp duplex part (T−G mismatches are also involved) was designed on either side of the nick. Mismatches are designed to promote later hybridization between two prepared circular ssDNAs. Concretely, a 92 nt long circular ssDNA (c92A) was obtained from two linear strands of l-92a (40 nt) and l-92b (52 nt), whose hairpin stems were 6 bp long. The stems for l-132a and l-132b (two fragments of 70 and 62 nt length to obtain the 132 nt long circular ssDNA of c132A) were 7 and 6 bp, respectively. After hybridization of these two fragments, two nicks were formed for circularization (see arrows in Figure 2B).

The electrophoresis analysis for circularization and removal of linear unreacted substrates are shown in Figure 2C. As we reported previously [31], to avoid polymeric byproducts, 0.05 × T4 DNA ligase buffer was used. As expected, within the concentration range (≤10 μM) we used, almost no substrates were left, demonstrating that the ligation was efficiently carried out (lanes 6−11, Figure 2C). The bands below the main circular products could be assigned to be linear strands obtained after only one nick was ligated. Significantly, polymeric products longer than 92 or 132 nt, which have been problematic previously, were not observed. As a result, the yield was higher than 80% (lanes 6−11, Figure 2C). After the cyclization products were digested by exonuclease I (Exo I, digesting linear ssDNA) and nuclease exonuclease III (Exo III, digesting linear dsDNA), relatively pure circular products of c92A and c132A were obtained (long linear byproducts were removed). The purity can be improved to almost 100% after purification by phenol treatment and ethanol precipitation removing ligases and other smaller molecules (data not shown).

### 2.2. Preparation of LR-Chimera Involving Partly an APP Sequence and an ssDNA Part for Extra Hybridization

One of our aims is to construct an LR-chimera (cc-91A) with an extra ssDNA part as a hanging hook to attach it to other molecules (Figure 1). As shown in Figure 3A, the sequence of the targeted LR-chimera contains an APP-rich part involving two continuous APP sequences of 24 and 8 bp to form stable Z-DNA (see the underlined part in Figure 3A). The ssDNA hook was 41 nt long, and the complementary parts between the two ssDNA circles were 91 nt long. After hybridization, an extra adenylate (dA) was present on c92A, and the extra 41-nt-long ssDNA part on c132A was used as the ssDNA hook. Actually, it is a challenge to hybridize these two circular ssDNAs (Figure 3), because our newly designed circular ssDNAs can form stable intra-molecular secondary structures, as we mentioned above (Figure 2B). When we just mixed c92A and c132A, no new band for the hybrid of cc-91A was observed, indicating that almost no hybridization occurred (data not shown).

Interestingly, after the oligonucleotide DNA of L59 (dark blue strand, Figure 3B) was added, which has a 41-nt-long complementary sequence (involving 2 mismatches) with the hook region on c132A, a new band with much lower mobility was observed (lane 9 in Figure 3C), indicating that c92A and c132A hybridized. The yield of hybridization was nearly 100% in a buffer with relatively low ionic strength (10 mM PBS, pH 7.2, 10 mM MgCl_2_), because no c132A was left. The new band was assigned as cc-L59-91A, which has lower mobility than B-cc-L59-91A (compare lane 9 with lane 11 in Figure 3C). B-cc-L59-91A is the B-form circular dsDNA of cc-91A (obtained by ligation after hybridization of linear l92A and circular c132A strand) hybridizing with L59. We further tried to remove L59 with L’49, which is partly complementary to L59. After L’49 was added to the above solution, two new bands were observed (lane 10 in Figure 3C). The band with higher mobility was assigned as the hybrid of L59 and L’49, and the lower one as cc-91A (the hybrid of c92A and c132A). The hybrid of cc-91A migrated slower than lc-91A and cl-91A, but faster than ll-91A (compare lane 10 with lanes 6−8, Figure 3C). The ll-91A, lc-91A, and cl-91A are hybrids of l92A (linear 92 nt strand) and l132A (linear 132 nt strand), l92A and c132A (circular 132 nt strand), and c92A (circular 92 nt strand) and l132A, respectively. As expected, cc-91A had similar mobility to B-cc-91A, which is B-form circular dsDNA (lane 12). Our previous results show that the LR-chimera usually has similar mobility as its topological isomer of B-form circular DNA [30,39]. Accordingly, it can be concluded that the LR-chimera of cc-91A involving an ssDNA hook was successfully prepared.

In our previous studies on LR-chimera [30,39], two circular ssDNAs were prepared with splints, and byproducts had to be removed with gel purification. In this study, the two ssDNAs to form cc-91A were circularized without the necessity of adding splints. This circularization method can achieve the preparation of circular ssDNA with high concentration (up to 10 μM) and high yield without polymeric byproducts. Although the stable intra-molecular secondary structure is not conducive to the hybridization of these two circular ssDNAs, the hybridization can be carried out efficiently with the aid of an oligonucleotide (L59) to hybridize to the loop. The aid oligonucleotide can be removed, and the formed LR-chimera is very stable.

### 2.3. Confirmation of the Z-DNA Part Present in cc-91A Using Z-DNA-Specific Binding Proteins

The Z22 antibody is usually used to prove the presence of Z-DNA because it can specifically recognize and bind to Z-DNA with high affinity and no sequence specificity [40,41]. Here, we also used Z22 to verify whether left-handed DNA was formed in cc-91A. The results of Z22 binding to cc-91A are shown in Figure 4A. With the increase of Z22 concentration, the band for cc-91A disappeared gradually, and new bands with lower mobility were observed, indicating that Z22 binds to cc-91A. Interestingly, when the Z22 concentration was as low as 0.25 μM (lane 3, Figure 4B), there was a clear product band (lane 3, Figure 4A), indicating that only one Z22 molecule bound to each cc-91A molecule. For 0.50 μM, another weak band was observed (lane 4), implying that two Z22 molecules may bind to cc-91A. When Z22 concentration increased to 0.75 μM or higher, the mobility of the complex of Z22 and cc-91A was almost zero, and bands were observed only in the well of gel (lanes 5−7), indicating that three or more Z22 molecules bound to cc-91A.

When ZBP1was used [42], gel shift was also observed (Figure 4B). In cases ZBP1 was 1.25 μM or lower (lanes 3−5, Figure 4B), only an intensity decrease for the cc-91A band was observed, indicating that ZBP1 was bound but not strong enough. During the gel shift assay, the dissociation of ZBP1 from cc-91A may occur to cause the smearing of product bands. When the concentration of ZBP1 was 1.875 μM or higher (lane 6 and lane 7), relatively clear bands could be observed but they were not so sharp. This result indicates that several ZPB1 molecules may occupy most positions of the left-hand part, and the remaining, free ZBP1 molecules cannot further bind to cc-91A. It should be noted that the ZBP1 molecule (MW = 19.4 KDa) is much smaller than Z22 (MW= 147.9 KDa). Obviously, Z22 and ZBP1 bind to Z-DNA with different patterns. It has been shown that Z22 may prefer to bind to Z-DNA close to B–Z junctions [17]. The LR-chimera we designed also contains a B–Z junction, so there may be two kinds of Z22 binding sites in the LR-chimera. When the concentration of Z22 is low, it may first bind to the B–Z junction. With the increase of concentration, other Z22 molecules began to bind to the Z-DNA part relatively far from the B–Z junction so that other bands with lower mobility formed. For ZBP1, no detailed binding information is available. As expected, no binding was observed under the same conditions for ll-91A and lc-91A (as negative controls), which cannot from Z-DNA (Figure 4C). Accordingly, we can conclude that cc-91A contains a Z-DNA part at the APP region.

### 2.4. Preparation and Characterization of APP-Poor LR-Chimera with an ssDNA Hook

In our recent study, we proved that a non-APP (non-alternating pyrimidine purine) sequence can also form stable left-handed DNA if the topological constraint is strong enough [39]. However, there is no report about binding affinity between Z-DNA of non-APP sequences and Z-DNA binding proteins. Here, we designed another sequence, cc-91B (Figure 5A), containing only two short APP sequences (5′-d(CATGCGTA)-3′, 5′-d(CATATG)-3′, underlined in Figure 5A) without d(CG)_n_ sequences. Obviously, cc-91B is an APP-poor sequence. Compared with the APP sequence, it is more difficult to form Z-DNA, because the difference in free energy between B-DNA and Z-DNA is bigger for a non-APP sequence compared with that of an APP one [39]. Considering that a stable secondary structure may inhibit the hybridization between two circular ssDNAs, we used the “one-pot method” with low MgCl_2_ concentration to circularize l92B and l132B (Appendix A) [31,34].

To obtain cc-91B (Figure 5A), we hybridized purified c92B and c132B (Figure 5B). As shown in Figure 5C, the bands for c92B and c132B almost disappeared, and a new band with lower mobility than ll-91B, lc-91B, and cl-91B was observed (lanes 6−9). The ll-91B, lc-91B, and cl-91B are controls of the hybrids of l92B and l132B, l92B and c132B, and c92B and l132B, respectively. Accordingly, the new band can be assigned as cc-91B, which should be an LR-chimera (lane 9, Figure 5C).

Z22 and ZBP1 proteins were also used to verify the presence of the Z-DNA duplex in cc-91B formed mainly by non-APP sequences (Figure 5D−F). With the increase of Z22 and ZBP1 concentration, substrate cc-91B decreased significantly (Figure 5D,E), but its B-DNA analogue (lc-91B) did not (Figure 5F), indicating that Z-DNA was present in cc-91B. As compared with the results for cc-91A (Figure 4), not-so-clear binding product bands were observed for cc-91B, implying that the binding to APP-poor sequences may be weaker than that to APP ones (Figure 5D,E). We speculate that the binding positions of Z22 and ZBP1 to cc-91B are more flexible. In other words, the B–Z junction may be different among various cc-91B molecules. This may also reflect that the binding ability of Z22 to the Z-DNA part of the non-APP sequence is weaker than that of the APP one. It should be noted that no MgCl_2_ was present in the gel, indicating again that Z-DNA can form at relatively low ionic strength. Accordingly, we have successfully prepared LR-chimeras with an ssDNA hook involving either an APP-rich sequence or an APP-poor sequence for the Z-DNA part. The LR-chimeras are stable under low ionic strength (e.g., 10 mM MgCl_2_, or 89 mM Tris-H_3_BO_4_ and 4 mM NaH_2_BO_4_). The ssDNA hook can be used to hybridize to another ssDNA strand to attach the LR-chimera to a surface for further analysis.

### 2.5. Evaluation of ZBP1 Binding Affinity to Non-Modified Z-DNA

A linear ssDNA attaching 5’-terminal biotin was used to immobilize the LR-chimera to the biosensor surface coating with streptavidin (Figure 6A). The hybrid complex of the LR-chimera and the biotin-modified ssDNA was designated as Bait-cc-91 (Bait-cc-91A and Bait-cc-91B for cc-91A and cc-91B, respectively). The hybridization for Bait preparation was simply carried out by mixing the corresponding LR-chimera and the biotin-modified ssDNA, and the controls of B-form DNAs (Bait-ll and Bait-lc) were also prepared (Appendix A). The affinity between Z-DNA and ZBP1 protein was evaluated using BLI by the protocol shown in Figure 6B. After loading (immobilization of Bait to the biosensor) and washing away unattached LR-chimera molecules, association (binding of ZBP1 to LR-chimera) and dissociation (dissociation of ZBP1 from LR-chimera) steps were carried out. The binding buffer contains 10 mM phosphate (pH 7.2), 10 mM MgCl_2_, 0.1% BSA, and 0.02% Tween-20 (BSA and Tween-20 avoid non-specific binding). The real-time binding curves of various concentrations of ZBP1 and LR-chimera are show in Figure 6C,D. The optimized Bait concentration was determined to be 20 nM, and the loading time was 240 s.

Figure 6C shows the time courses for binding of various concentrations of ZBP1 to mobilized cc-91A involving APP-rich sequence (solid lines). The shape (pattern) of these binding curves is quite typical, and the binding height was as high as about 2.5 nm (for 400 nM ZBP1), which is also ideal for dynamic analysis. After fitting the measurement results with a binding model (dotted lines), the corresponding parameters were obtained. The association rate K_on_, dissociation rate K_off_, and affinity constant K_D_ (K_off_/K_on_) were determined to be 8.28 × 10^3^ M^−1^s^−1^, 4.68 × 10^−6^ s^−1^, and 0.566 nM, respectively. The R^2^ for fitting was as high as 0.9996. Accordingly, the binding of Z-DNA-specific proteins to Z-DNA could be investigated by using the prepared Bait at various conditions.

The binding of ZBP1 to cc-91B (APP-poor sequence) was also investigated (Figure 6D). Even when 400 nM ZBP1 was used, the binding height was only about 0.15 nm, which is only about 1/16 of that for cc-91A under the same conditions (comparing Figure 6D with Figure 6C). Our attempt at fitting failed, and the affinity constant for ZBP1 binding to cc-91B could not be obtained. When B-DNA (lc-91A) was used instead of cc-91B, the binding height was only about 0.06 nm, indicating that almost no binding was observed (Figure 6E). As a result, ZBP1 can bind to non-APP (or APP-poor) sequence very weakly, at least in the presence of 10 mM MgCl_2_. This is also consistent with the electrophoresis result of ZBP1 binding to cc-91B (Figure 5E). We can conclude from these results that ZBP1 has strong sequence specificity: the binding strength for the APP sequence we used in this study was much stronger than that for non-APP sequences, although we have shown that these sequences can also form a stable left-handed duplex [39]. As shown in Figure 3A, cc-91A involves two parts of continuous APP sequences, and one of them contains a 6-nt-long 5′-d(CGCGCG)-3′). Detailed analysis is required to obtain the information explaining of this difference. The analysis of the binding of other proteins to various sequences of Z-DNA is underway in our lab.

As we know, an extremely high salt concentration (e.g., 1.0 M MgCl_2_) is usually necessary for the stabilization of non-modified linear Z-DNA in vitro, at which ZBP1 or other proteins are easily denatured. For our approach, the circular LR-chimera can form stable Z-DNA at low salt concentration, which is an important prerequisite for a ZBP1 binding assay. Therefore, for the first time, the binding curves of unmodified Z-DNA with both APP-rich and APP-poor sequences to ZBP1 were obtained in the presence of 10 mM MgCl_2_. This strategy makes it easy to study the binding dynamics of Z-DNA of unmodified arbitrary sequences with various proteins or other molecules, and the investigation of ionic effects becomes much more convenient.

## 3. Conclusions

We successfully constructed an LR-chimera with an ssDNA loop for hybridization, which can attach it to other molecules. The Z-DNA duplex was also formed even in the presence of this extra ssDNA part. In addition, we developed an efficient ssDNA circularization method by the simple hybridization of two fragments followed by ligation, and splints were not essential. We showed that the affinity of ZBP1 binding to non-modified Z-DNA can be investigated simply by the hybridization of a biotin-modified DNA to the constructed LR-chimera (Bait-cc91). The binding constant of ZBP1 to Z-DNA was also obtained (0.566 nM) in the presence of 10 MgCl_2_. For the first time, we showed that ZBP1 had weak binding to non-APP sequences of Z-DNA even when the left-handed duplex was stable enough. We believe that this method is facile and can be used for the quantitative analysis of Z-DNA as well as Z-RNA binding under various conditions.

## 4. Materials and Methods

### 4.1. Materials

T4 DNA ligase, exonuclease I, and exonuclease III were purchased from Thermo Scientific (Pittsburgh, PA, USA). The Ultra GelRed (a dye staining both dsDNA and ssDNA) was purchased from Vazyme (Nanjing, China). The Z-DNA-specific antibody (Z22) was from Absolute Antibody Ltd. (Oxford, UK), and ZBP1 (recombinant Z-DNA-binding protein) was from Wuhan USCN Business Co., Ltd. (Wuhan, China). The streptavidin (SA) biosensor was purchased from SAIDELISI (Tianjin, China). All other chemicals were from Sigma-Aldrich (St. Louis, MO, USA).

All oligonucleotides used in this study (see Appendix A for sequences) were purchased from GENEWIZ (Suzhou, China). Sequences were designated as follows. l: linear ssDNA strand; c: circular ssDNA strand; Sp: directed the head–tail ligation of a linear oligonucleotide to a circular one; lc and cl: hybridization of a linear strand and a circular one; cc: hybridization of two circular strands, B-cc: T4 DNA ligase sealed cl or lc.

### 4.2. Preparation of Circular ssDNA

For the preparation of c92A, 5.0 μL of linear ssDNA l-92a (40 μM) and l-92b (40 μM), and 2.0 µL of 0.5 × T4 DNA ligase buffer (20 mM Tris-HCl, 5 mM MgCl_2_, 5 mM DTT, 0.25 mM ATP, pH 7.8 at 25 °C) were added into 7.0 µL H_2_O. After being heated to 90 °C for 3 min, the solution was cooled gradually (0.1 °C/s) to 37 °C and kept for 20 min. Then, 1.0 µL T4 DNA ligase (5 U) was added. The ligation reaction (20 µL in total) was carried out at 37 °C for 2 h. The ligase was deactivated by incubation at 65 °C for 10 min. The preparation of c132A was the same as above. For the preparation of c92B, 1 μL of linear ssDNA l_n_-92a (40 μM) and l_n_-92b (40 μM), 2 µL of 1 × T4 DNA ligase buffer (40 mM Tris-HCl, 10 mM MgCl_2_, 10 mM DTT, 0.5 mM ATP, pH 7.8 at 25 °C), and 3 µL of Sp_n_92a (40 μM) and Sp_n_92b (40 μM) were added into 9 µL H_2_O. After being heated to 90 °C for 3 min, the solution was cooled gradually (0.1 °C/s) to 37 °C and kept for 20 min. Then, 1 µL T4 DNA ligase (5 U) was added. The ligation reaction (20 µL in total) was carried out at 37 °C for 2 h. The preparation of c132B used 1 × T4 DNA ligase buffer (final concentration), and other operations were the same as for c92B.

To the above reaction containing the circular ssDNA solution (20 µL), 1.0 µL of 10 × exonuclease I reaction buffer (670 mM glycine-KOH, 67 mM MgCl_2_, 10 mM DTT, pH 9.5 at 25 °C), 1.0 µL of 10 × exonuclease III reaction buffer (660 mM Tris-HCl, 6.6 mM MgCl_2_, pH 8.0 at 30 °C), 1.0 µL of exonuclease I (20 U), and 0.5 µL of exonuclease III (20 U) were added, then the digestion reaction was carried out at 37 °C for 12 h. Exonuclease I and III were deactivated at 80 °C for 15 min. The proteins and ions in the above solution were removed with phenol/chloroform and ethanol. Finally, the obtained circular DNA samples were stored in the buffer of 0.1 × TE (1.0 mM Tris-HCl and 0.1 mM EDTA, pH 8.0) and analyzed using a Nanodrop 2000 spectrophotometer (Thermo Scientific, Waltham, MA, USA).

### 4.3. Preparation of LR-Chimera with an ssDNA Part

For cc-91A, 2.0 μL of c92A (10 μM), 2.0 μL of c132A (10 μM), 2.0 μL of L59 (10 μM), and 2 μL of 10 × PBS buffer containing MgCl_2_ (100 mM NaH_2_PO_4_, pH 7.2, 100 mM MgCl_2_) were added into 9.6 μL H_2_O. After being heated to 90 °C for 3 min, the solution was cooled gradually (0.1 °C/s) to 50 °C and kept for 2 h, and then continued to cool to 25 °C and was kept for 20 min. Finally, 2.4 µL of linear ssDNA L’40 (10 μM) was added to the above solution and incubated at 25 °C for 12 h (20 µL in total). For cc-91B, 2.0 μL of c92B (10 μM), 2.0 μL of c132B (10 μM), and 2 μL of 10 × PBS buffer (containing MgCl_2_) were added into 12.0 μL H_2_O. After being heated to 90 °C for 3 min, the solution was cooled gradually (0.1 °C/s) to 25 °C and kept for 20 min (20 µL in total). LR-chimeras with an ssDNA part were analyzed by 8% PAGE at 20 °C.

### 4.4. Preparation of Bait

For a 20 μL reaction system, 4.0 μL of c92 (5 μM), 4.0 μL of c132 (5 μM), 4.0 μL of L_B_50 (5 μM), 2.0 μL of 10 × PBS buffer (containing MgCl_2_), and 6.0 μL of H_2_O were mixed. The solution was heated at 90 °C for 3 min, gradually cooled (0.1 °C/s) to 50 °C and kept for 2 h, and then continued to cool to 25 °C and was kept for 20 min. The product was assigned as Bait-cc-91 (containing left-handed DNA). For Bait-cl containing only B-DNA, 4.0 μL of l92 (5 μM), 4.0 μL of c132 (5 μM), 4.0 μL of L_B_50 (5 μM), 2.0 μL of 10 × PBS buffer (containing MgCl_2_), and 6.0 μL of H_2_O were mixed. The solution was heated at 90 °C for 3 min and gradually cooled (0.1 °C/s) to 25 °C and kept for 20 min. All products were analyzed by 6% PAGE at 20 °C.

### 4.5. Evaluation of Affinity between ZBP1 and Z-DNA

The affinity of ZBP1 protein binding to the non-modified Z-DNA was evaluated using BLI and an Octet Red 96 system (Molecular Devices, San Jose, CA, USA) at 25 °C. The assay procedure included five steps: baseline (60 s); loading (180 s); baseline 2 (180 s); association (480 s); and dissociation (480 s). Z-DNA was 20 nM; ZBP1 were 25, 50, 100, 200, and 400 nM. The buffer used contained 1 × PBS buffer (containing 10 mM MgCl_2_) (pH 7.2), 0.1% BSA, and 0.02% Tween-20. The response data and affinity parameter (K_D_) were obtained using the Octet Data Analysis Software Version 11.1.

## Figures and Tables

**Figure 1 molecules-27-03706-f001:**
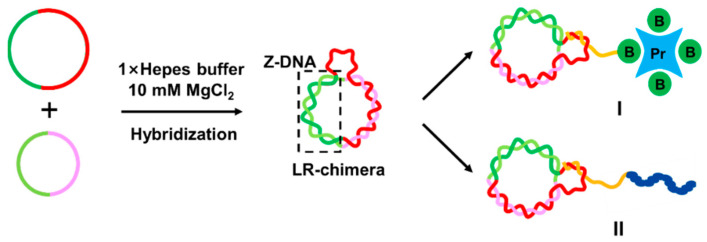
Schematic diagram of LR-chimera involving non-modified Z-DNA attaching to a variety of molecules. This molecular design has many applications. For example, LR-chimera can hybridize to an ssDNA attaching a biotin group, which binds strongly to streptavidin (I), or directly hybridize to the ssDNA part of a chimera involving another polymer chain of peptides or polysaccharides (II).

**Figure 2 molecules-27-03706-f002:**
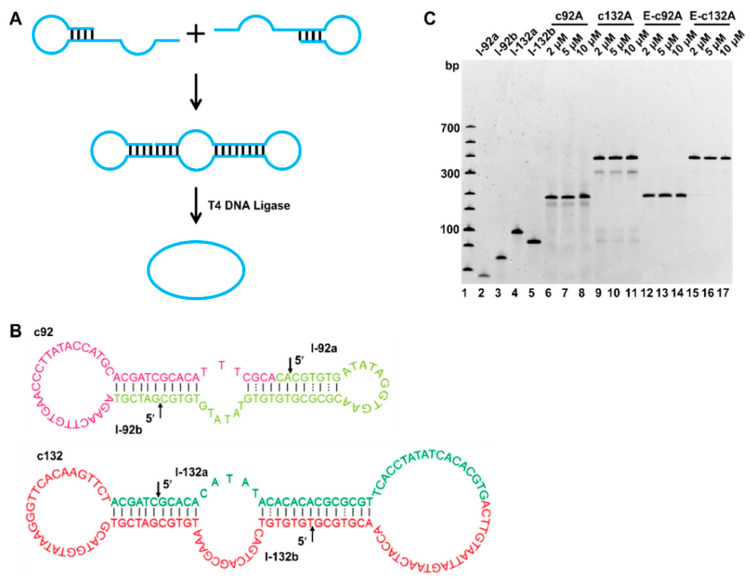
Preparation and sequence design for efficient cyclization of ssDNAs. (**A**) Schematic diagram for circularization without splints. T4 DNA ligase sealed two nicks formed by the hybridization of two fragments. (**B**) Sequence design for c92A and c132A. Red letters show the part forming B-DNA (mainly of non-APP sequences) and green ones (APP-rich sequences) are to form Z-DNA. (**C**) Cyclization results analyzed by 6% dPAGE. Lanes 2−5: linear ssDNA fragments of l-92a (40 nt), l-92b (52 nt), l-132a (70 nt), and l-132b (62 nt), respectively; lanes 6−8: ligation product (c92A) analysis at various DNA concentrations (2, 5, and 10 μM); lanes 9−11: ligation products (c132A) analysis at various DNA concentrations; lanes 12−14: samples shown in lanes 6−8 were digested by exonuclease I (Exo I) and exonuclease III (Exo III); lanes 15−17: samples shown in lanes 9−11 were digested by Exo I and Exo III. Other conditions: [T4 DNA ligase buffer] = 0.05 ×, [T4 DNA ligase] = 0.25 U/μL, 37 °C, 2 h.

**Figure 3 molecules-27-03706-f003:**
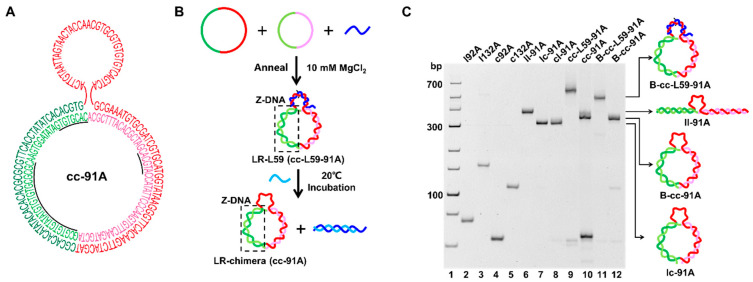
Preparation of LR-chimera cc-91A. (**A**) Sequence of cc-91A with the extra ssDNA part. Letters with underlines show the continuous APP sequences. (**B**) Schematic diagram for process of cc-91A preparation. A part of L59 (dark blue) is complementary to the hook region on c132A (involving two mismatches for its quick replacement by L’49). L’49 (light blue) is completely complementary to a part of L59. (**C**) PAGE (8%) analysis of the hybridization of various strands. Lanes 2−5: l92A, l132A, c92A, and c132A, respectively; lanes 6−8: hybrids of l92A and l132A (ll-91A), l92A and c132A (lc-91A), and c92A and l132A (cl-91A); lane 9: hybrid of c92A, c132A, and L59 (cc-L59-91A); lane 10: L’49 was added to the sample shown in lane 9 (cc-91A and a short duplex about 49 bp long were formed); lane 11: hybrid of L59 with the B-form circular dsDNA (B-cc-91A, obtained by ligation after hybridization of l92A and c132A) to form B-cc-L59-91A; lane 12: B-cc-91A. Hybridization conditions: [phosphate] = 10 mM, pH 7.2, [MgCl_2_] = 10 mM, [linear ssDNA] = 1.0 μM, [circular ssDNA] = 1.0 μM. The gel was kept at 20 °C during electrophoresis.

**Figure 4 molecules-27-03706-f004:**
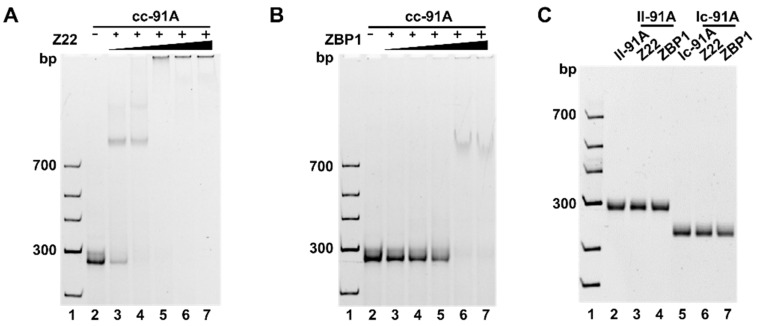
Gel shift assay for binding of Z22 (Z-DNA-specific antibody) and ZBP1 to cc-91A (6% PAGE). (**A**) Z22. Lanes 3−7: 0.25, 0.5, 0.75, 1.0, and 1.25 μM of Z22. (**B**) ZBP1. Lanes 3−7: 0.25, 0.625, 1.25, 1.875, and 2.5 μM of ZBP1. (**C**) Controls for Z22 and ZBP1 binding to B-DNA. Lanes 3 and 6: 1.25 μM of Z22; lanes 4 and 7: 2.5 μM of ZBP1. Other conditions: 0.25 μM of cc-91A, ll-91A, or lc-91A. Buffer: [phosphate] = 10 mM, pH 7.2, [MgCl_2_] = 10 mM. All gels were kept at 20 °C during electrophoresis.

**Figure 5 molecules-27-03706-f005:**
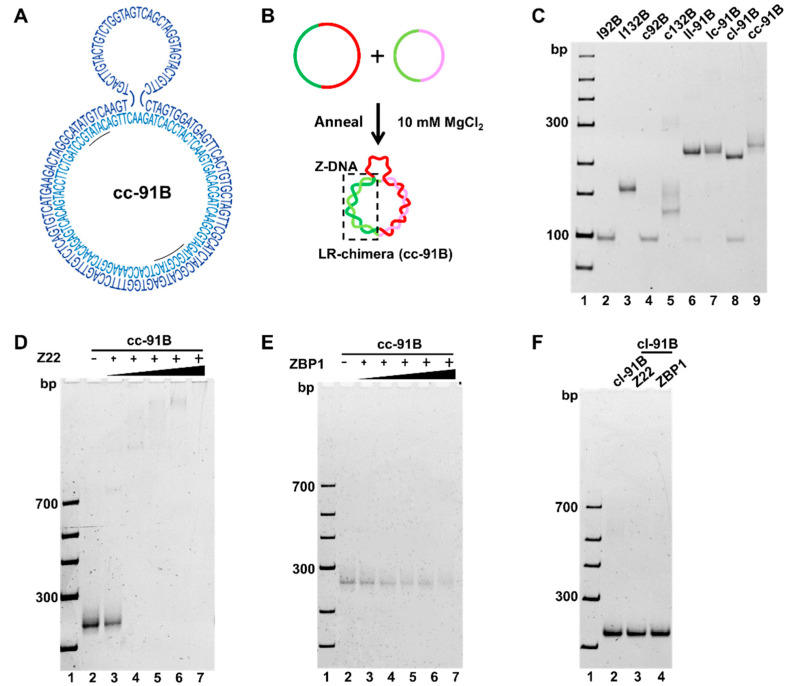
Preparation of cc-91B and gel-shift assay after binding by Z22 and ZBP1. (**A**) Sequence design. The underlined letters show a continuous APP sequence. (**B**) Scheme for cc-91B preparation by hybridization of c92B and c132B. (**C**) Electrophoresis analysis of cc-91B preparation (8% PAGE). Lanes 2−5: l92B, l132B, c92B, and c132B, respectively; lanes 6−8: hybrids of l92B and l132B (ll-91B), l92B and c132B (lc-91B), and c92B and l132B (cl-91B), respectively; lane 9: hybrid of c92B and c132B (cc-91B). (**D**) Z22 binding to cc-91B. Lanes 3−7: 0.25, 0.5, 0.75, 1.0, and 1.25 μM of Z22, respectively. (**E**) ZBP1 binding to cc-91B. Lanes 3−7: 0.25, 0.625, 1.25, 1.875, and 2.5 μM of ZBP1, respectively. (**F**) Controls for Z22 and ZBP1 binding to B-DNA (lc-91B). Lane 2: lc-91B only; lane 3: 1.25 μM Z22; lane 4: 2.5 μM ZBP1. For (D−F), 0.25 μM cc-91B or 0.25 μM lc-91B was used. Buffer: [phosphate] = 10 mM, pH 7.2, [MgCl_2_] = 10 mM. All the gels were kept at 20 °C during electrophoresis.

**Figure 6 molecules-27-03706-f006:**
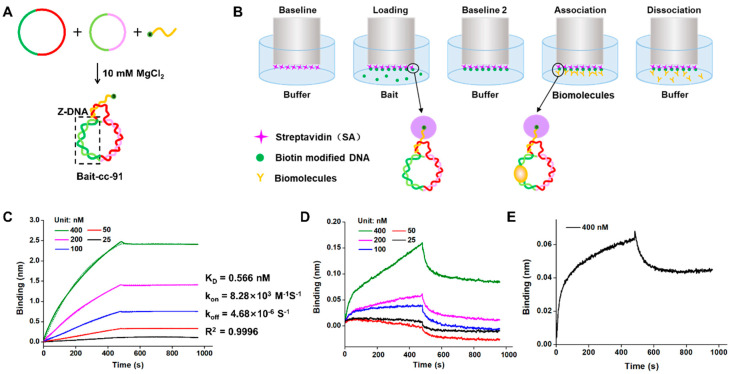
Evaluation of the affinity of non-modified Z-DNA and ZBP1 protein. (**A**) Preparation of Bait-cc-91. (**B**) Process (from left to right) for evaluation of the affinity between Z-DNA and other biomolecules. (**C**−**E**) Response curves for ZBP1 binding to cc-91A, cc-91B, and B-DNA, respectively. Solid lines: measured data; dotted lines: fitting curve. Conditions: [phosphate] = 10 mM, pH 7.2, [MgCl_2_] = 10 mM; [Bait] = 20 nM; baseline: 60 s; loading time: 240 s; baseline 2: 180 s; association time: 480 s; dissociation time: 480 s.

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
