# Peer review of "Construction of ssDNA-Attached LR-Chimera Involving Z-DNA for ZBP1 Binding Analysis"

_molecules, 2022, doi:10.3390/molecules27123706_

Round 1
Reviewer 1 Report
The author proposes contraction of Z-DNA containing LR-chimera, which is formed from two ssDNAs of different sizes. The resulted circular DNA contained both left-handed and right-handed DNA. Different sizes of ssDNA used for chimera construction, and this provides for remaining ssDNA that can be further attached to biotin group or peptide or polysaccharide. This construct helps to investigate Z-DNA binding under different. conditions.
The verification of Z-DNA formation was done with two proteins ZBP1 and Z22. The results showed different patterns of binding Z22 and ZBP1 where several molecules of Z22 can bind Z-DNA with the increase of Z22 concentration while only one molecule ZBP1 can bind the same Z-DNA length. I suggest authors discuss why ZBP1 molecule, which is much smaller (MW = 19.4 KDa) than Z22 (MW= 147.9 KDa) can bind only one copy to the constructed Z-DNA while 2 or 3 copies of Z22 can bind the same Z-DNA.
The authors also investigated the affinity of binding depending on sequence specificity - whether the sequences are alternating purine-pyrimidine (APP) or not. These are very interesting results, showing that ZBP1 prefers APP sequences though it does bind for some period of time to non-APP sequence. It should be stated more clearly in the text whether only (CG)n were tested or other APP-seqneces too.
It would be interesting to see the same binding curves for Z22.
Overall I think that the paper is important and provides an experimental background to study Z-DNA binding properties at different conditions and with different proteins.
Reviewer 2 Report
The is an extension based on their recent JACS publication (10.1021/jacs.8b13855). However, the only difference seems to be introduction of an additional sequence for hybridization with a biotin-containing ssDNA, so that the approach can be applied to bio-layer interferometry (and other techniques, e.g., SPR, requiring immobilization of the DNA construct). Thus, the overall advance seems to be rather incremental. Obviously, if the author could present an example where BLI has revealed new insight about Z-DNA, which could not be studied by other approaches, it would make this a strong manuscript with clear scientific advances.
Reviewer 3 Report
The reviewed manuscript describes a novel biochemical way to construct the Z-DNA which is left-handed and very unstable. They also found the double-stranded Z-DNA together with mixed left-handed and the right-handed (LR-chimera) Z-DNA and single-strand DNA (ssDNA). It is an interesting study in the biochemistry field. The biggest problem with this manuscript is its language. There are a number of grammar errors and vague expressions, which significantly affect the normal reading and understanding of the manuscript. Specific abbreviations are not properly introduced when it appears for the First time in the abstract and main context. Besides, I have several other questions or suggestions:
1). How stable is the prepared Z-DNA? Does it need a specific condition to keep its stability?
2). The authors have published several other ways of forming Z-DNA in vitro. What is the pro and con of the current methods compared to others (for instance, Zhang et al. JACS 2019; Li et al. NAR 2022)? It will be good if the authors could add these useful discussions in a discussion section.
3). In the abstract and main context line 67, the authors mentioned “splints”. Related background introduction is missing.
Round 2
Reviewer 2 Report
The authors provided explanations to my question. However, I am confused what they meant that it was "a big challenge because ZBP1 as well as other proteins are hard to use at high concentration of salts." Does this mean that their method can stablize these proteins at high salt concentrations? Or their method enables formation of stable Z-DNA at low salt concentration? Or something else? In any cases, the authors should include their argument into the Discussion to highlight the merit of this manuscript/work.
Reviewer 3 Report
My technical questions have been addressed. The English problem is partially solved. However, the overall writing is still not good enough for publication. The authors should find a native speaker to read it through and make positive modifications. For example, the most important session, the abstract reads badly.
